# Physics-Informed Neural Networks for Thermo-Responsive Hydrogel Swelling: Integrating Constitutive Models with Sparse Experimental Data

**DOI:** 10.3390/ma18235401

**Published:** 2025-11-30

**Authors:** Seyed Amirmasoud Takmili, Eunsoo Choi, Alireza Ostadrahimi, Mostafa Baghani

**Affiliations:** 1School of Mechanical Engineering, College of Engineering, University of Tehran, Tehran 14399-57131, Iran; 2Faculty of Mechanical Engineering, K.N. Toosi University of Technology, Tehran 19919-43344, Iran; 3Department of Civil and Environmental Engineering, Hongik University, Seoul 04066, Republic of Korea

**Keywords:** physics-informed neural networks, thermo-responsive hydrogels, free swelling, unidirectional constrained swelling, gel collapse temperature

## Abstract

Poly(N-isopropylacrylamide) (PNIPAM) hydrogels are temperature-sensitive materials whose swelling is difficult to predict near the gel collapse temperature (GCT). A physics-informed neural network (PINN) was developed in which a stabilized free-energy model is embedded and sparse data for free and uniaxially constrained swelling are assimilated. Across datasets, the PINN reduces test RMSE by 44% at low crosslink density (Nν = 0.0035) and by 65% at higher density (Nν = 0.02), with coverage inside ±RMSE bands improving from 61.9% to 76.2% for free swelling at Nν = 0.0035. Under constraint, test relative error decreases from 19.95% to 11.86% (n = 1) and from 9.19% to 5.90% (n = 3), while preserving thermodynamic stability. The method sharpens the transition slope near the gel-collapse temperature and narrows prediction intervals without overfitting, capturing cold-regime plateaus and hot-side tails more faithfully. The framework integrates governing equations with data to deliver accurate swelling and stress predictions using only eight anchors and thirteen held-out points per case. These results position PINNs as a reliable surrogate for designing thermo-responsive hydrogels in soft actuators and biomedical systems.

## 1. Introduction

In soft matter systems, poly(N-isopropylacrylamide) (PNIPAM) hydrogels are well-known for their abrupt swelling and deswelling upon temperature change [1]. This creates both chances for new uses and problems for predictive modeling [2,3,4]. This behavior has garnered considerable interest in biomedical applications, particularly in hydrogels utilized for injectable drug delivery systems that facilitate localized and regulated release without invasive procedures [5,6,7,8]. Their adjustable swelling and deswelling in tissue engineering helps encapsulate cells and produce dynamic scaffolds that more closely resemble natural extracellular environments [9,10,11,12]. Thermo-responsive networks are being researched more and more for moisture capture and release [13,14,15,16]. For example, interpenetrating PNIPAM-alginate structures can absorb water from the air at lower temperatures and release it when they are slightly heated. This makes them an efficient way to collect water sustainably [17,18]. Furthermore, rapid swelling-induced volume alterations form the foundation for soft actuators and microrobots that undergo reversible shape transformations in response to thermal stimuli [19,20,21]. Even with these improvements, practical use is still limited by mechanical weakness, slow response times caused by diffusion limits, and small operational windows around the gel collapse temperature [22,23]. These flaws show how important it is to create predictive modeling methods that can reliably capture swelling behavior and help design hydrogel systems for a wide range of engineering uses [24,25,26,27].

For a long time, theoretical models of hydrogel swelling have used free-energy formulations that combine the elastic deformation of the polymer network with the mixing of the polymer and solvent. Most of the models that came after the classical Flory–Rehner framework were based on it. It made rough predictions about equilibrium swelling but did not account for the sharp changes that happen near the gel collapse temperature [28,29]. To overcome these constraints, Cai and Suo [30] enhanced the formulation by integrating a composition- and temperature-dependent interaction parameter, facilitating improved concordance with experimental data at moderate cross-link densities. Their model, on the other hand, has multiple equilibrium solutions close to the swelling, which causes snap-through instabilities that make numerical implementation more difficult and make it impossible to use reliably in finite element simulations. Mazaheri et al. [31] built on this idea by proposing a modified free-energy model that used a polynomial approximation instead of the logarithmic term that caused instability. This got rid of multiple-solution behavior and made sure that the transition region was stable [31]. This formulation successfully avoids discontinuities and produces continuous equilibrium predictions across a broad spectrum of conditions; however, it tends to underestimate swelling at low cross-link densities and under constrained deformation. This suggests that physics-only models are inadequate for fully capturing all experimentally observed trends.

Even though there have been considerable improvements in constitutive modeling, it is still hard to get both thermodynamic stability and quantitative accuracy in different experimental settings. Stabilized free-energy formulations forecast persistent equilibrium yet still underestimate swelling in networks characterized by low cross-link density and in geometrically constrained scenarios. Models based only on data also have problems because hydrogel experiments are often costly, take a long time, and give only sparse or noisy measurements. These restrictions have led to the creation of hybrid methods that mix physics-based modeling with data integration. Physics-informed neural networks are a very promising framework because they put governing equations directly into the learning process and use experimental data to fix systematic errors at the same time. Their efficacy has been validated in various domains of mechanics, including fracture dynamics in solids [32,33], finite-strain plasticity modeling [34,35], inverse identification of constitutive laws under large deformation [36], and calibration of hyperelastic behavior from full-field measurements [37]. The ability of Physics-Informed Neural Networks (PINNs) to overcome data scarcity while preserving physical consistency in computational mechanics has been further highlighted by recent studies [38]. According to these studies, when traditional physics-only or purely data-based methods fall short, PINNs can act as reliable stand-ins. Because stability of the free-energy landscape near the gel collapse temperature must be maintained and data-informed correction is needed to capture experimentally observed deviations, applying this paradigm to thermo-responsive hydrogels is especially promising.

In this work, we create a neural network framework for thermo-responsive hydrogels based on physics, which combines the corrective power of sparse experimental data with the stability of a free-energy formulation. The method relies on established constitutive laws, like the polynomial-based model of Mazaheri, Baghani and Naghdabadi [31], which addresses systematic departures from experimental observations and guarantees continuous swelling predictions throughout the transition region. A two-stage training strategy is employed: initial physics-based pretraining to guarantee equilibrium consistency and a single-well free-energy landscape, followed by refinement with experimental data to improve quantitative accuracy. This methodology is consistent with recent advances in physics-informed learning where neural networks embed constitutive structure to preserve stability [39,40] while incorporating data to correct residual bias. In order to assess predictive performance in comparison to both experimental data and physics-only models, the framework is tested against representative swelling scenarios documented in the hydrogel literature [41,42,43].

The remainder of this paper is organized as follows. Section 2 presents the theoretical foundations and network architecture. Section 3 describes the training methodology, datasets, and evaluation metrics. Section 4 reports the results and discusses their implications relative to existing models. Section 4 concludes with a summary of the contributions, limitations of the present approach, and perspectives for future research.

## 2. Theory

The equilibrium states of PNIPAM hydrogels can be distinguished by their swelling behavior under different thermal and mechanical conditions (Figure 1). At high temperature, the polymer network undergoes collapse, which defines the reference configuration. When cooled below the transition temperature, the gel absorbs water and swells isotropically, resulting in a free-swelling state (Figure 1a). If axial deformation is restricted by rigid boundaries, swelling becomes uniaxially constrained, and the gel expands only in the lateral direction (Figure 1b). These distinct states form the basis for the continuum formulation presented below.

### 2.1. Governing Equations of Hydrogel Swelling

The mechanical response of thermo-responsive hydrogels in equilibrium can be formulated within the framework of finite-strain continuum mechanics [44]. Consider a representative volume element of a dry hydrogel network in the reference state, characterized by material coordinates X. Under thermal and mechanical loading, the network deforms to the current state with coordinates x(X), and the deformation gradient is defined as(1)$$F=\frac{\partial x}{\partial X},$$
where F captures local volume changes and distortions of the polymer network [31]. The associated right Cauchy–Green tensor is given by [45](2)$$C=F^{T}F.$$

To describe equilibrium swelling, the total free energy density per unit reference volume is expressed as a combination of elastic stretching and polymer–solvent mixing contributions. Denoting the Helmholtz free energy density by W, one can write [29]:(3)$$W=W_{\mathrm{stretch}}(F,T)+W_{\mathrm{mix}}(c,T),$$
where $W_{\mathrm{stretch}}$ accounts for the entropy-driven elasticity of the polymer chains, $W_{\mathrm{mix}}$ captures the thermodynamic mixing between polymer and solvent molecules, c is the solvent concentration, and T is the absolute temperature [30].

The hydrogel volume change originates from solvent uptake, which can be expressed through the swelling constraint [46]. Let ν denote the molecular volume of solvent and J=detF the volume ratio. The absorbed solvent content is then related to the deformation by(4)J=1+νc,
a constraint that couples network deformation to solvent diffusion [29]. This condition can be incorporated into the free energy using a Lagrange multiplier, leading to augmented potential:(5)$$W=W_{\mathrm{stretch}}(F,T)+W_{\mathrm{mix}}(c,T)+P(J-1-\nu c),$$
where P is the Lagrange multiplier associated with incompressibility of the polymer–solvent system [31].

For the stretching contribution, a neo-Hookean representation is commonly adopted, which for polymer networks can be written as:(6)$$W_{\mathrm{stretch}}=\frac{1}{2}Nk_{B}T(I_{1}-3-2lnJ),$$
where N is the effective number of polymer chains per unit reference volume, $k_{B}$ is the Boltzmann constant, and $I_{1}=tr(C)$ is the first invariant of C. This form reflects the entropic elasticity of the polymer chains while accounting for finite volume change through the logarithmic term [30].

The mixing contribution is based on Flory–Huggins theory, in which the interaction parameter χ depends on both temperature and polymer volume fraction. In the classical model, the mixing free energy is expressed as:(7)$$W_{\mathrm{mix}}=\frac{k_{B}T}{\nu }[(J-1)ln(\frac{J-1}{J})+\chi \frac{J-1}{J}],$$
where the interaction parameter χ reflects the enthalpic contribution of polymer–solvent interactions. For PNIPAM hydrogels, experiments show that χ depends both on temperature and polymer volume fraction. A convenient form, adopted by Cai and Suo [30] and later by Mazaheri, Baghani and Naghdabadi [31], expresses it as:(8)$$\chi =\chi_{0}+f\chi_{1},\;\chi_{0}=A_{0}+B_{0}T,\;\chi_{1}=A_{1}+B_{1}T,$$
where f=1/J is the polymer volume fraction, and $A_{0}$, $B_{0}$, $A_{1}$, $B_{1}$ are material constants calibrated against experimental swelling data [47].

Although this model reproduces many swelling behaviors, the logarithmic term in $W_{\mathrm{mix}}$ generates multiple equilibrium states near the gel collapse temperature, producing snap-through instabilities that hinder numerical implementation [30]. To eliminate this problem, Mazaheri, Baghani and Naghdabadi [31] replaced the logarithmic term with a truncated series expansion, retaining the first three terms. The resulting modified free energy becomes:(9)$$W=\frac{1}{2}Nk_{B}T(I_{1}-3-2lnJ)+\frac{k_{B}T}{\nu }(J-1)(-\frac{1}{J}-\frac{1}{2J^{2}}-\frac{1}{3J^{3}})+\frac{k_{B}T}{n}(\frac{\chi_{0}}{J}+\frac{\chi_{1}}{J^{2}}).$$

This polynomial-based modification ensures continuity of the free-energy landscape and guarantees the existence of a unique equilibrium state across the gel collapse region.

Equilibrium swelling is achieved by minimizing the total free energy with respect to the relevant stretches, subject to the swelling constraint. In the case of constrained swelling, the reaction stress is computed through the first Piola–Kirchhoff stress, which is given by the axial component. The non-dimensional reaction stress is expressed as:(10)$$P_{z}=N\nu (\lambda_{fix}-\frac{1}{\lambda_{fix}})+(-\frac{1}{J}-\frac{1}{2J^{2}}-\frac{1}{3J^{3}})+\frac{\chi_{0}-\chi_{1}}{J}+\frac{2\chi_{1}}{J^{2}}$$

The stress is expressed in non-dimensional units, scaled by $k_{B}T/\nu$. The material parameters listed in Table 1 correspond to those reported by Cai and Suo [30], who modeled hydrogel swelling experiments from previous studies [41,42,43,48]. The present ANL-only (stabilized) model follows the Cai–Suo formulation with the logarithmic regularization proposed by Mazaheri, Baghani and Naghdabadi [31] to enhance numerical stability near the gel collapse temperature. The training and testing datasets used here originate from these experimental systems to ensure direct comparability and model validation.

This formulation completes the constitutive description required for the subsequent physics-informed neural network framework.

### 2.2. Physics-Informed Neural Network Framework

To integrate the constitutive model described in Section 2.1 with experimental observations while preserving thermodynamic consistency, a Physics-Informed Neural Network (PINN) framework is employed. In this context, the term “neural network” refers to a machine learning approach that combines classical thermodynamic principles with data-driven refinement. This framework embeds the governing equations derived from the free energy and its equilibrium conditions directly into the training process through automatic differentiation. By incorporating physical laws into the learning objective, the model ensures that predictions remain consistent with thermodynamic constraints, while also being informed by sparse experimental data. This approach enables the model to extrapolate reliably, particularly in regions where experimental data may be limited.

The network is constructed to map thermodynamic drivers to equilibrium deformations. For free swelling, the input is temperature and the output is the isotropic stretch ratio. For unidirectional swelling, the inputs are temperature and the prescribed longitudinal stretch, while the output is the corresponding transverse stretch. A fully connected feed-forward architecture with smooth activation functions is adopted, which guarantees differentiability of the outputs with respect to the inputs and enables direct evaluation of the equilibrium conditions through automatic differentiation.

Equilibrium is imposed through the stationarity of free energy. In the free swelling case, the residual is defined as(11)$$R_{\mathrm{free}}(T)=\frac{\partial W}{\partial J}(J(T),T),$$
while for unidirectional swelling, where one stretch is fixed and the other two are equal, the residual becomes(12)$$R_{\mathrm{uni}}(T,\lambda_{\mathrm{fix}})=\frac{\partial W}{\partial \lambda_{t}}(\lambda_{t}(T,\lambda_{\mathrm{fix}}),\lambda_{\mathrm{fix}},T),\;\;J=\lambda_{\mathrm{fix}}\lambda_{t}^{2}.$$

The physics-based contribution to the loss, $L_{phys}$, is defined as the squared residual of Equations (11) or (12), depending on the swelling mode. Here, angular brackets ⟨⋅⟩ denote averaging over collocation points in the temperature domain and, in the uniaxial case, over prescribed stretches. Experimental observations are assimilated using a robust Huber loss,(13)$$L_{\mathrm{data}}=\langle {Huber}_{\delta }(\lambda_{\mathrm{pred}}(T)-\lambda_{\exp}(T))\rangle +\langle {Huber}_{\delta }(\lambda_{t,\mathrm{pred}}(T,\lambda_{\mathrm{fix}})-\lambda_{t,\exp}(T,\lambda_{\mathrm{fix}}))\rangle ,$$
where “pred” and “exp” refer to predicted and measured stretches, respectively; the first term is used for free swelling, and the second for uniaxial swelling.

To enhance stability and reflect thermodynamic constraints, additional regularizers are introduced. Convexity of the free-energy landscape is promoted by penalizing negative curvature in the volume ratio,(14)$$L_{conv}=\langle ReLU(-\frac{\partial^{2}W}{\partial J^{2}}{)}^{2}\rangle ,$$
which suppresses spurious non-convex wells. Thermo-responsive contraction is enforced by constraining the temperature derivative of the stretch,(15)$$L_{mono}=\langle ReLU(\frac{\partial \lambda }{\partial T}{)}^{2}\rangle ,$$
so that the network does not predict expansion with increasing temperature. Weak value and slope priors relative to a calibrated baseline are also employed,(16)$$L_{prior}=\beta_{v}\langle w_{out}(T)(\lambda -\lambda_{base}{)}^{2}\rangle +\beta_{s}\langle w_{out}(T)(\frac{\partial \lambda }{\partial T}-\frac{\partial \lambda_{base}}{\partial T}{)}^{2}\rangle ,$$
acting mainly outside the GCT window, with $w_{out}(T)$ a smooth indicator that vanishes inside the transition region. The complete training objective is therefore(17)$$L_{total}=\alpha L_{phys}+L_{conv}+\gamma L_{mono}+L_{prior}+w_{data}L_{data}.$$
with adaptive scheduling of the weights to balance physics, regularization, and data assimilation during training.

To exclude unphysical compression and maintain smooth gradients, the volume ratio is parameterized with a “softplus” barrier,(18)J(T)=1+softplus(z(T)),
where z(T) is the scalar network output. This guarantees J ≥ 1 by construction and matches the implementation used in training. For free swelling, the isotropic stretch is recovered from *J* as(19)$$\lambda (T)=J(T{)}^{1/3}$$

This mapping is evaluated by automatic differentiation for all temperature derivatives used in the loss.

For unidirectional swelling with a prescribed longitudinal stretch $\lambda_{fix\;}=\;n\;\lambda_{0}$ (with integer n and reference $\lambda_{0}$), the transverse stretch used for training is(20)$$\lambda_{t}(T,\lambda_{fix})=\sqrt{\frac{J(T,\lambda_{fix})}{n\lambda_{0}}}$$

The network outputs z(T), which is mapped to J via Equation (18) and then to $\lambda_{t}$ through Equation (20).

Training is carried out with the full objective function from the outset, combining physics residuals, regularizers, and data misfit. To prevent data terms from dominating early, their weights are ramped gradually during optimization. This continuous scheme eliminates the need for a staged protocol while still ensuring thermodynamic consistency and stable assimilation of sparse experimental data [40,49]. The coefficients used for the different contributions to the training objective are summarized in Table 2.

All model training and evaluation were performed on a workstation equipped with an Intel^®^ Core™ i7-12700H CPU (Intel Corp., Santa Clara, CA, USA), 16 GB RAM, and an NVIDIA RTX 4070 Laptop GPU (8 GB VRAM; NVIDIA Corp., Santa Clara, CA, USA), running Windows 11 Pro and Python version 3.10. The implementation employed TensorFlow version 2.15 (tf.keras deep-learning API included in TensorFlow) with NumPy version 1.26 and pandas version 2.2. Optimization was conducted using the Adam algorithm (initial learning rate $10^{-3}$) with gradient-norm clipping (∥g∥≤1) and a step-decay learning-rate schedule. Each configuration was trained for 6000 epochs with a fixed random seed (1234) to ensure reproducibility. Training required approximately 5–6 min per dataset (≈2 min per 2500 epochs).

The PINNs for both free and uniaxial swelling are fully connected feed-forward networks with three hidden layers, 64 neurons per layer, and tanh activations to ensure smooth temperature derivatives. Weights were initialized with Glorot uniform initialization. The data term uses a Huber loss with δ=0.03, combined with physics residuals and regularizers as defined in Equations (11)–(16). Training employed batches of 256 collocation points per iteration; the data-misfit weight was ramped during training to avoid early dominance of data terms while maintaining thermodynamic consistency. The same architecture and schedule were used in both geometries to ensure consistent capacity and convergence behavior.

## 3. Results and Discussion

This section compares an analytical thermodynamic baseline (ANL-only, stabilized) with a physics-informed extension (ANL–PINN). The ANL-only (stabilized) model implements classical gel free-energy with a numerically stabilized formulation (log-form/regularized terms) to ensure robust solutions near the collapse region, and it is solved with standard finite-element methods. The ANL–PINN augments the same thermodynamic structure by training a neural correction term against experimental data while enforcing equilibrium and energy consistency, thereby retaining physical constraints. We evaluate both approaches for swelling under uniaxial constraint and free swelling.

In the case of experimental data, only a limited number of measurements were employed to demonstrate the data efficiency of the PINN approach. Specifically, eight anchor points were selected and evenly distributed across the cold, near-transition, and hot regimes to capture key physical features while avoiding overfitting. The remaining points were reserved exclusively for performance evaluation.

### 3.1. Energy Landscapes

We first examine how constitutive energy varies with temperature and mechanical constraint, because the depth of the local minimum and its curvature govern branch selection and numerical sensitivity near the transition. Figure 2 reports model-only landscapes—no experimental points or learned outputs are used. Classical analyses show that PNIPAM gels can exhibit multiple stationary states [50] or very low curvature near the GCT [51], which complicates standard minimization and often requires specialized branch-tracking; stabilized variants were proposed to mitigate these issues [52].

In the free-swelling cases (Figure 2a,b), the 307 K curves exhibit a single minimum with $J^{*}$ near ~2.2 and moderate curvature. At 300 K the minimum shifts to larger swelling and the curvature drops markedly, indicating a locally flat landscape below the transition. When curvature is small, the energetic restoring force is weak and nearby states become nearly degenerate, so small thermal or numerical perturbations can switch the selected branch [31].

Under uniaxial constraint (Figure 2c,d), the 307 K minima again cluster near $J^{*}\approx 2$ with curvature comparable to free swelling. At 300 K the same shift-and-flattening is amplified by constraint, and increasing n further weakens curvature, thereby intensifying branch fragility and step-size sensitivity.

Overall, Figure 2a–d show that the numerically delicate regime is the low-curvature minimum near 300 K—not a deep well at 307 K—motivating careful branch tracking/regularization and targeted calibration of curvature where solvers are least reliable.

### 3.2. Free Swelling

Free-swelling responses were evaluated for three crosslink densities using sparse temperature anchors for training and withholding the remaining points for testing; results are shown separately in Figure 3 (Nν=0.0035), Figure 4 (Nν=0.01), and Figure 5 (Nν=0.02). In each case, the proposed PINN (blue curve with its ±RMSE band) was contrasted with the stabilized constitutive prediction (black curve with its ±RMSE band). The comparison was designed to probe the two known trouble spots of thermo-responsive gels—namely, the sharp drop near the transition and the bias that tends to accumulate in the colder regime where the energy landscape is locally flat—while keeping supervision deliberately sparse.

In Figure 3 (Nν=0.0035), the transition was matched more closely by the PINN, and the constitutive model’s cold-side underestimation was reduced; the blue curve tracks the low-temperature plateau at the level of the measurements, whereas the analytical baseline sits below them. On held-out temperatures, test RMSE decreased from 0.151 to 0.085, with MAE from 0.132 to 0.051 and MAPE from 8.28% to 3.17%. Coverage within each method’s own ±RMSE band increased from 61.9% (baseline) to 76.2% (PINN). These gains were achieved without over-tightening: the PINN band is narrower but continues to span the steep portion of the knee, indicating that curvature was controlled while avoiding oscillations under sparse anchoring.

At the intermediate density (Figure 4, Nν=0.01), differences are smaller but systematic: test RMSE is 0.108 for PINN–ANL versus 0.113 for the reference, MAPE is 4.56% versus 6.52%, and the share of test points inside each method’s own ±RMSE band is 70.8% versus 62.5%; the gain is concentrated on the warm side, above 308 K, where the PINN–ANL better tracks the measured tail, while through the knee between 300 and 308 K, both curves follow similar trajectories, and on the cold side, below 300 K, a small positive signed error remains at two held-out points, indicating a mild over-prediction that does not affect overall stability.

At the highest density (Figure 5, Nν =0.02), the strongest contrast appears in two windows. First, across the transition (about 306–310 K), the slope of the PINN–ANL curve is nearly twice that of the analytical reference (about −0.128 per K versus −0.068 per K), which yields a sharper knee consistent with the measurements and underlies the reduction in aggregate error (test RMSE 0.032 vs. 0.092; MAE 0.019 vs. 0.085; MAPE 1.40% vs. 7.09%; inside-band coverage 80% vs. 60%). Second, in the cold regime (about 290–298 K), a plateau is observed in the measurements and is reproduced by the PINN–ANL with a small magnitude slope (about −0.007 per K), whereas the analytical reference decays more quickly (about −0.035 per K) and therefore crosses from slight over-prediction at the very cold end to under-prediction closer to the knee. This combination—flatter pre-transition behavior and a steeper, data-aligned knee—explains both the lower residuals and the higher fraction of points within the ±RMSE band for the PINN–ANL. Physically, the near-plateau reflects weak temperature sensitivity of the effective interaction in this range together with the higher elastic penalty at larger Nν that damps volumetric response until the cooperative collapse is triggered; once the transition is approached, the response becomes dominated by the interaction change, producing the sharper descent captured by the PINN–ANL.

Taken together, Figure 3, Figure 4 and Figure 5 show that the analytical reference deviates both through the transition and away from it—underestimating the cold plateau at higher crosslink density and relaxing the warm tail at intermediate density—whereas the PINN–ANL maintains the low-temperature plateau, sharpens the drop, and better aligns the high-temperature tail under sparse supervision; the reductions in test RMSE are ≈44% at Nν =0.0035 and 65% at Nν =0.02 (with a measurable gain at Nν=0.01), and these improvements are accompanied by narrower ±RMSE bands with higher test coverage, indicating genuine error reduction rather than optimistic banding, so embedding equilibrium consistency and smoothness in the loss and calibrating with a few temperature anchors offers a robust path to recover the GCT behavior and correct tail biases without introducing spurious features or instability.

### 3.3. Constrained Swelling

Building on the free-swelling results, the uniaxial constrained cases in Figure 6, Figure 7 and Figure 8 demonstrate the same trend: the analytical model captures the overall shape but smooths the transition and misrepresents the extremes, while the PINN-ANL adjusts toward the experimental response around the GCT and preserves stable behavior where data are sparse.

As shown in Figure 6, eight sparse anchors and thirteen held-out points were used for the uniaxial case at Nν=0.01 and $\lambda_{fix\;}=2\lambda_{0}$. In this configuration, the PINN–ANL reduced the test relative error from 19.95% to 11.86%, with RMSE decreasing from 0.224 to 0.182, MAE from 0.194 to 0.114, and MAPE from 20.24% to 12.81%. Test-point coverage within the ±RMSE band improved from 71.4% for the analytical reference to 85.7% for the PINN–ANL. The sharp drop recorded between 306 K and 308 K remains challenging for both models: the analytical reference produces a softened descent, while the PINN–ANL shifts closer to the experimental branch without fully reaching the depth of the measurements. In the colder regime, the distinction between the two approaches becomes clearer. Where anchors or test data are absent, the PINN–ANL relaxes toward the analytical plateau, reflecting the stabilizing effect of the embedded physics. However, in regions where measurements exist, the PINN–ANL departs from the analytical prediction and adjusts toward the observed values, thereby reconciling the physical regularization with the experimental diversity. This selective deviation—anchored by sparse data but constrained by thermomechanical consistency—demonstrates the key advantage of the approach: it reproduces the underlying physics while adapting to experimental sharpness in ways that a purely analytical model cannot.

In Figure 7, the uniaxial case with $\lambda_{fix\;}=\;2\lambda_{0}$ shows the most consistent agreement of the PINN–ANL with experiment. Where anchors are absent, the PINN–ANL collapses to the analytical baseline, since only the physics-informed constraint acts, but in regions with data, the PINN–ANL departs from the analytical curve and aligns with the measurements. The analytical model overestimates the cold response and underestimates the sharp GCT drop, while the PINN–ANL overlaps both deviations and removes any systematic bias. Beyond 308 K, the analytical prediction drifts gradually into its plateau, whereas the PINN–ANL approaches the plateau more abruptly, consistent with the observed lateral stretch. This sharpening is also evident in the metrics: RMSE decreases from 0.146 to 0.069, MAE from 0.102 to 0.043, and MAPE from 7.69% to 3.71%, with the maximum deviation halved (0.334 to 0.172). The case illustrates how the framework preserves the analytical baseline away from anchors, while exploiting sparse data to adjust curvature and remove biases in the critical transition region.

In the strongest constrained case, reported in Figure 8, the PINN–ANL again outperformed the analytical reference, though the improvement was more modest than at $\lambda_{fix\;}=\;2\lambda_{0}$. The test relative error decreased from 9.19% to 5.90%, MAPE from 10.42% to 7.34%, and inside-band coverage increased from 77.8% to 83.3%. RMSE remained close between the two approaches (0.113 vs. 0.100), while a single sharp deviation raised the maximum error for the PINN–ANL (0.337 vs. 0.228). In the cold region the analytical curve tends to overestimate, and the PINN–ANL reduces this offset more effectively than the other cases, while near the GCT both curves underestimate the steep experimental drop, but the PINN–ANL remains closer overall. Around 307 K, two underestimations are visible, yet they remain localized and do not affect the broader trend. Most notably, beyond 308 K the PINN–ANL reaches the plateau cleanly and follows the hot-side branch with minimal lag, unlike the analytical model, which approaches it more gradually. Overall, the case at n = 3 demonstrates that the PINN–ANL maintains proportional agreement across the range, improves plateau behavior, and reduces systematic deviations, with residual discrepancies confined to isolated points.

### 3.4. Reaction Stress Under Uniaxial Constraint

The non-dimensional first Piola stresses derived under uniaxial constraint are shown in Figure 9, and they reveal a strong dependence on temperature and constraint level. At the transition temperature near 310 K, the PINN–ANL consistently produces larger compressive stresses than the analytical reference, with increases of roughly 30–45% across all constraint levels. This amplification reflects the sharper collapse in swelling captured by the PINN–ANL, which translates directly into a stronger mechanical response under loading.

Away from the transition, the relative ordering of the two models can invert and the differences shrink. In the colder regime (300–305 K), the analytical baseline is nearly flat across constraint levels (change ≈ 0.002–0.003), indicating a plateau, whereas the PINN–ANL relaxes more over the same interval (Δ ≈ 0.0146 for all n). On the hot side at 315 K, the analytical model approaches the plateau more gradually and remains around 7–9% stronger than the PINN–ANL, consistent with the lag it shows in reaching the post-transition branch of the swelling curve. In the post-transition regime, the PINN–ANL relaxes as the collapsed plateau is reached and the constraint-induced mismatch diminishes.

The influence of constraint is explicit: at low temperatures, stronger constraint (larger n) reduces the magnitude of the compressive reaction by accommodating more swelling axially; near the GCT, the same constraint amplifies the peak compression due to a larger mismatch between the constrained and free-swelling states, while on the hot side its effect diminishes as the response approaches the collapsed plateau.

Overall, the comparison highlights that the largest discrepancies occur at the GCT, where the physics-informed strategy sharpens the stress response in line with the steep experimental drop, while at the temperature extremes the two models converge and occasionally exchange order. This behavior demonstrates that the PINN–ANL does not simply rescale the analytical prediction but redistributes stress magnitudes in a way that is physically consistent with both constraint and the experimentally observed swelling transition. Because direct reaction-stress measurements for these constrained protocols are not available in the cited literature, Figure 9 should be interpreted as a comparison of model behaviors rather than an assessment of accuracy.

## 4. Conclusions

The results presented across free and constrained swelling, together with the reaction stress analysis, establish several consistent trends. The energy landscape analysis showed that the most numerically delicate regime is not the sharp minimum at the GCT but the flat curvature at lower temperatures, especially when constraint is increased. This explains why purely analytical models, even when stabilized, are prone to bias or branch sensitivity.

In free swelling, the PINN–ANL demonstrated its ability to preserve stability while adjusting toward the experimental response with only sparse temperature anchors. At low crosslink density, it corrected the cold-side underestimation; at intermediate density, it improved the warm-side tail; and at high density, it sharpened the transition slope while reproducing the observed cold plateau. These effects translated into substantial reductions in RMSE, MAE, and MAPE, with improvements of up to 65% in test error, accompanied by narrower uncertainty bands and higher test coverage.

Under uniaxial constraint, the same pattern persisted: the analytical baseline captured the overall shape but smoothed the transition and lagged in approaching plateaus, while the PINN–ANL departed from it where anchors were available and moved closer to the experimental measurements. The gains were most evident at $\lambda_{fix\;}=\;2\lambda_{0}$, where systematic bias was eliminated and errors halved, while at $\lambda_{fix\;}=\;\lambda_{0}$ and $\lambda_{fix\;}=\;3\lambda_{0}$ the improvements were smaller but still clear in plateau tracking and inside-band coverage. Importantly, where no experimental anchors existed, the PINN–ANL converged to the analytical baseline, showing that the method inherits the stabilizing features of the physics yet remains capable of deviating when guided by data.

The reaction stress comparison reinforced these findings. The PINN–ANL magnified compressive stresses at the GCT by 30–45% relative to the analytical baseline, consistent with its sharper prediction of the swelling collapse, while away from the transition the two models were closer and sometimes exchanged order. This redistribution of stresses indicates that the physics-informed strategy captures not only the volumetric response but also the associated load transfer under constraint.

Taken together, these results demonstrate that embedding equilibrium consistency and smoothness into the loss, and calibrating with only sparse data, provides a robust pathway to capture the sharp GCT transition, correct systematic deviations in both swelling and stress, and preserve stability in regimes where conventional solvers or purely analytical formulations are least reliable.

## Figures and Tables

**Figure 1 materials-18-05401-f001:**
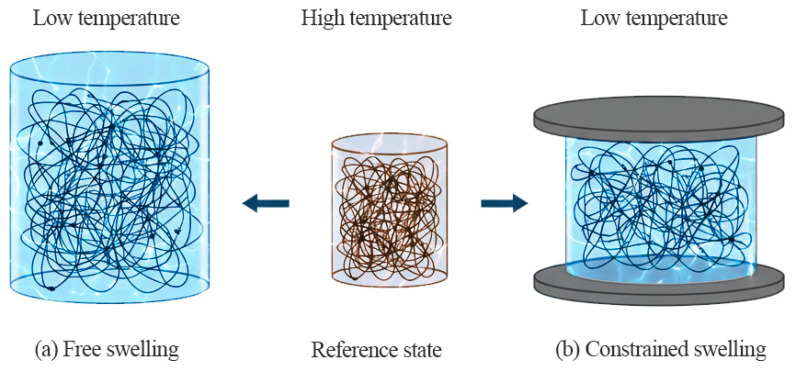
Representative states of PNIPAM hydrogel: reference, free swelling, and uniaxially constrained swelling.

**Figure 2 materials-18-05401-f002:**
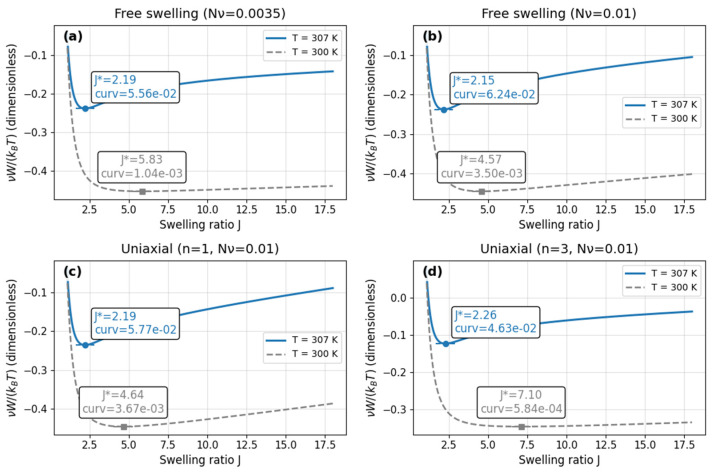
Model-only constitutive energy landscapes at 300 K (dashed) and 307 K (solid) for (**a**) free, Nv=0.0035; (**b**) free, Nv=0.01; (**c**) uniaxial, n=1, Nv=0.01; (**d**) uniaxial, n=3, Nv=0.01—each shows a single minimum, with 300 K shifting to larger $J^{*}$ and lower curvature, most pronounced under stronger constraint.

**Figure 3 materials-18-05401-f003:**
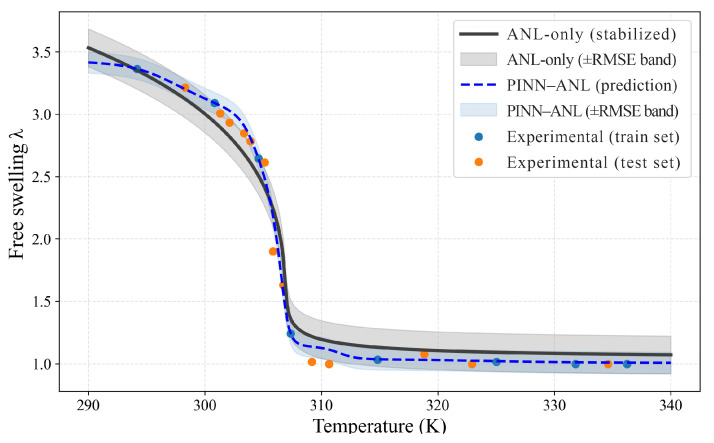
Free swelling, Nν=0.0035 PINN–ANL (blue dashed) and ANL-only (black solid) with ±RMSE bands and training/testing measurements.

**Figure 4 materials-18-05401-f004:**
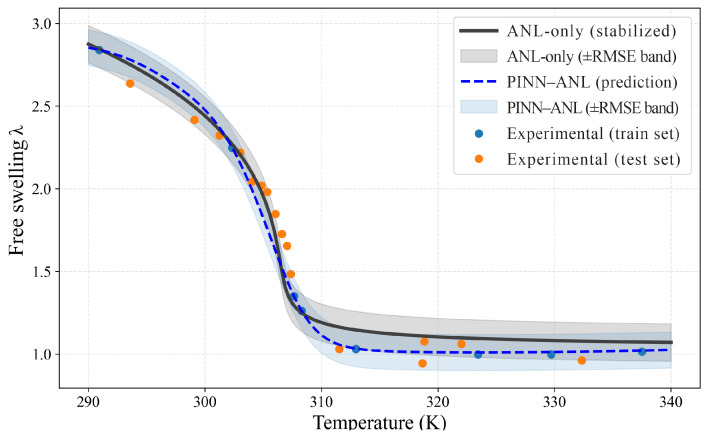
Free swelling, Nν =0.01 PINN–ANL (blue dashed) and ANL-only (black solid) with ±RMSE bands and training/testing measurements.

**Figure 5 materials-18-05401-f005:**
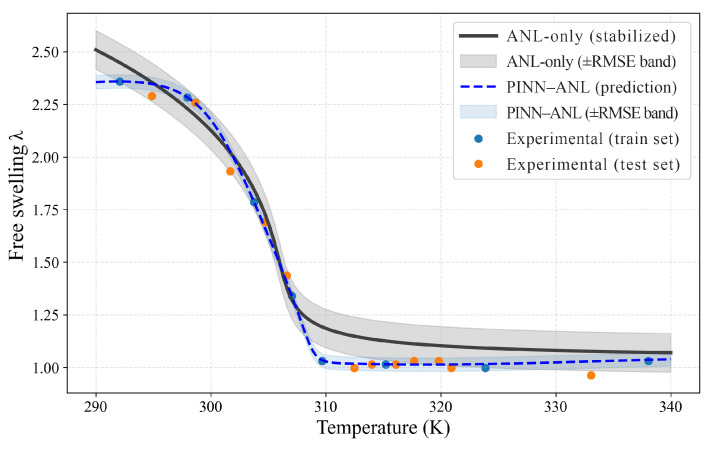
Free swelling, Nν =0.02 PINN–ANL (blue dashed) and ANL-only (black solid) with ±RMSE bands and training/testing measurements.

**Figure 6 materials-18-05401-f006:**
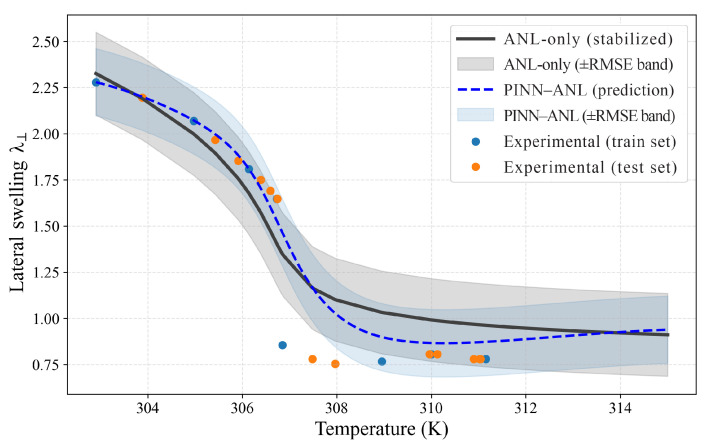
Swelling under uniaxial constraint with Nν=0.01 and fixed axial stretch $\lambda_{fix\;}=\lambda_{0}$, reporting lateral swelling $\lambda_{⊥}$ as a function of temperature with ±RMSE bands and training/testing measurements.

**Figure 7 materials-18-05401-f007:**
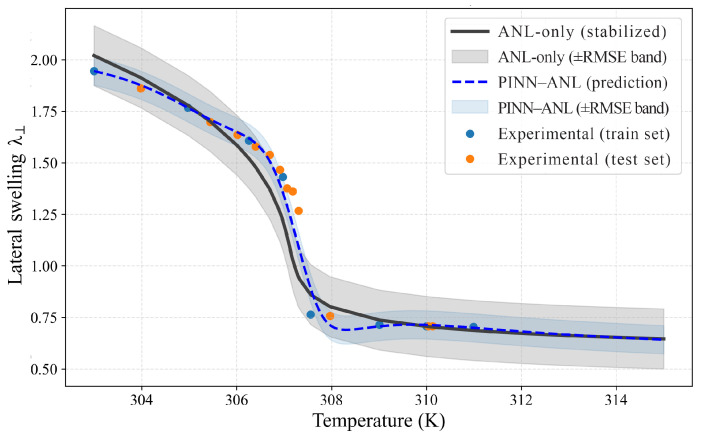
Swelling under uniaxial constraint with Nν=0.01 and fixed axial stretch $\lambda_{fix\;}=\;2\lambda_{0}$, reporting lateral swelling $\lambda_{⊥}$ as a function of temperature with ±RMSE bands and training/testing measurements.

**Figure 8 materials-18-05401-f008:**
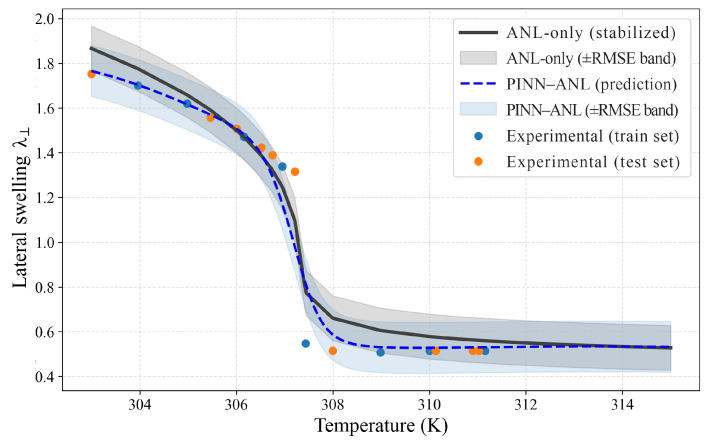
Swelling under uniaxial constraint with Nν=0.01 and fixed axial stretch $\lambda_{fix\;}=\;3\lambda_{0}$, reporting lateral swelling $\lambda_{⊥}$ as a function of temperature with ±RMSE bands and training/testing measurements.

**Figure 9 materials-18-05401-f009:**
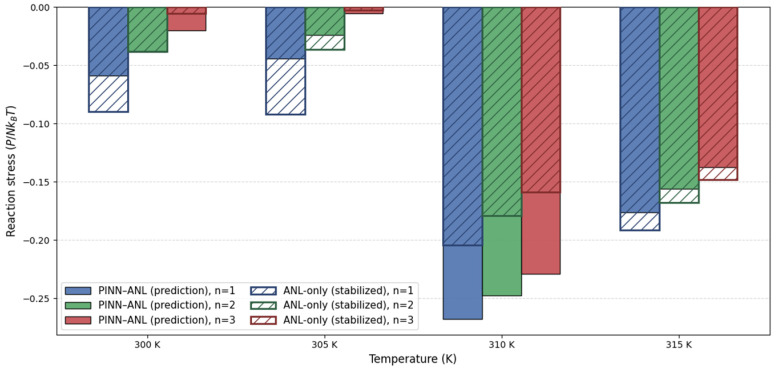
First Piola stress at 300, 305, 310, and 315 K for n=1, 2, 3 of uniaxially constrained swelling.

**Table 1 materials-18-05401-t001:** Material parameters for PNIPAM hydrogels used in the constitutive model.

Quantity	Free Swelling	Uniaxial Swelling	Description
A0	−12.947	−12.947	Temperature-independent component of χ
B0	0.04496	0.04496	Temperature-dependent component of χ
A1	17.92	17.92	Volume-fraction dependent component of χ
B1	−0.0569	−0.0569	Temperature-dependent component of χ via volume fraction
Reference stretch	−	1.737	Baseline longitudinal stretch used in uniaxial swelling

**Table 2 materials-18-05401-t002:** Training coefficients for the PINN framework in free and uniaxial swelling.

Loss-Term Weight ^1^	Free Swelling	Uniaxial Swelling	Description
Physics residual weight	0.01	1	Contribution of equilibrium residuals
Monotonicity penalty weight	0.3	0.3	Penalizes non-contractile response with temperature
Value prior weight	0.003	0.003	Regularizes deviation from baseline stretch outside GCT
Slope prior weight	0.002	0.002 (reduced ×0.30 inside GCT)	Regularizes slope relative to baseline outside GCT
Data loss weight (final target)	8.0 (ramped)	6.0 (ramped)	Weight applied to Huber misfit against experiments

^1^ Modest reweighting around the nominal values shows that a higher data weight sharpens the GCT knee but slightly increases tail bias, whereas a higher physics weight smooths the knee and improves extrapolation in sparse regions; monotonicity/convexity suppress local non-monotone and secondary-well artifacts; value/slope priors mainly stabilize behavior outside the transition. Within ±30–50% of nominal values, qualitative trends and conclusions are unchanged.

## Data Availability

The experimental swelling data used in this study were digitized from previously published PNIPAM curves and are provided as a single archive in the Appendix A containing the curated CSV datasets for free swelling and uniaxial constraint. The original sources did not report uncertainty; therefore, the digitized values are used as-is (no artificial noise and no outlier removal). Further inquiries can be directed to the corresponding author. For dataset sources, see Appendix A, which specifies that all constrained-swelling data derive from Suzuki (1997) [41], free-swelling data derive from Oh et al. (1998) [43] and Suzuki (1999) [48], and that the free-swelling subset at crosslink density Nν =0.02 reflects a legacy mixed case (Oh/Suzuki).

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
