# Peer review of "Physics-Informed Neural Networks for Thermo-Responsive Hydrogel Swelling: Integrating Constitutive Models with Sparse Experimental Data"

_materials, 2025, doi:10.3390/ma18235401_

Round 1

Reviewer 1 Report

Comments and Suggestions for Authors

Takmili et al. seek to model PNIPAM swelling behaviors, particularly the critical points near the phase transition, by combining the physics law with neutral networks. PINN is a well-established method that has been applied in many important fields. Overall, this work is timely and meaningful; however, many issues need to be addressed to ensure reproducibility.

Major concerns:

  1. Please elaborate how the model was trained. Specify the computational setting, including the hardware, software environment, and total training time. Indicate what code and framework were used, and share the code in a public repository or as Supporting Information.
  2. Collect all the experimental data as a dataset and share it with an open link or in the Supporting Information. Clarify the source, accuracy, and preprocessing steps and explain how it was generated and whether it included realistic noise.
  3. Explain all the hyperparameters used for NN, such as the number of hidden layers and neurons per layer, activation functions, learning rate, optimizer, number of epochs, batch size, and random seed for initialization, and many others.
  4. What influence does different weighting of loss terms on the final results. At least add a discussion about it.

Minor concerns:

  1. In the introduction, LCST should be correctly defined as lower critical solution temperature, not as the phase transition temperature. The abbreviation must correspond to its extended form.
  2. Format all the variables in the main text. All of them should be italicized. The authors made some italicized, while some not, such as Nv.

Author Response

Please find in attached.

Reviewer 2 Report

Comments and Suggestions for Authors

The manuscript entitled, ‘Physics-Informed Neural Networks for Thermo-Responsive Hydrogel Swelling: Integrating Constitutive Models with Sparse Experimental Data’ reported stimuli responsive hydrogel and their physical attributes. The article should be modified according the following comments:

  1. The abstract lacks specificity regarding the data presented in the study. It is recommended to highlight key findings or notable data to give readers a clearer understanding of the study's contributions.
  2. The choice of “eight anchors and thirteen held-out points” needs justification—how generalizable is this setup?
  3. The reversal of model performance away from the transition needs explanation; what mechanisms cause the analytical model to outperform in colder or hotter regimes?
  4. The discussion would benefit from explicit mention of how constraint level influences these stress trends.
  5. The explanation of curvature trends is clear but overly descriptive summarizing the key physical takeaway (why low curvature affects branch selection) would strengthen impact.
  6. Some articles would be significance for your reference:
  • Ganguly, S., Das, P., Srinivasan, S., Rajabzadeh, A. R., Tang, X. S., & Margel, S. (2024). Superparamagnetic amine-functionalized maghemite nanoparticles as a thixotropy promoter for hydrogels and magnetic field-driven diffusion-controlled drug release. ACS Applied Nano Materials7(5), 5272-5286.
  • Yin, B., Gosecka, M., Bodaghi, M., Crespy, D., Youssef, G., Dodda, J. M., ... & Zolfagharian, A. (2024). Engineering multifunctional dynamic hydrogel for biomedical and tissue regenerative applications. Chemical Engineering Journal487, 150403.

Author Response

Please find in attached.

Reviewer 3 Report

Comments and Suggestions for Authors

This study attempts and claims modelling of the swelling behavior, including the gel collapse as well, for free swelling and under uniaxial stress of a thermoresponsive polymer gel by the so called “Physics-Informed Neural Network” (PINN) based on sparse data. It is proposed to accept this paper for publication after some revision on the basis of comments below.

COMMENTS

1.

It is absolutely misleading to use the “neural network” expression in this case. The authors should explain in detail what they mean “neural network” on the basis of the well-known thermodynamic approach of gel collapse for crosslinked thermoresponsive polymers.

2.

The authors write that their method outperforms conventional methods (line 26). What kind of conventional methods the authors have in mind? In order to verify this claim, the authors have to carry out calculations by the “conventional methods” and compare their results with data obtained by modelling with “conventional methods”.

3.

The source of the “Experimental” (train set and test set) data are not provided, not explained at all. This should be included.

4.

The authors mention several times LCST as term for thermoresponsive polymer chains. This is wrong. A crosslinked thermoresponsive polymer has gel collapse and reswelling, but not LCST, and more broadly, shrinking-swelling, deswelling-reswelling, dehydration-rehydration. For instance, see and cite in the References section of the manuscript at least the following publications:

Zhang, X. Z.; Yang, Y. Y.; Wang, F. J.; Chung, T. S. Thermosensitive poly(N-isopropylacrylamide-co-acrylic acid) hydrogels with expanded network structures and improved oscillating swelling−deswelling properties. Langmuir 2002, 18, 2013-2018.

https://doi.org/10.1021/la011325b

Kato, N.; Sakai, Y.; Shibata, S. Wide-Range Control of Deswelling Time for Thermosensitive Poly(N-isopropylacrylamide) Gel Treated by Freeze-Drying.

Macromolecules 2003, 36, 961-963.

https://doi.org/10.1021/ma0214198

Osvath, Z.; Toth, T.; Ivan, B. Sustained Drug Release by Thermoresponsive Sol–Gel Hybrid Hydrogels of Poly(N‐Isopropylacrylamide‐co‐3‐(Trimethoxysilyl) Propyl Methacrylate) Copolymers. Macromol. Rapid Commun. 2017, 38, 1600724.

https://doi.org/10.1002/marc.201600724

Nishizawa, Y.; Inui, T.; Namioka, R.; Uchihashi, T.; Watanabe, T.; Suzuki, D. Clarification of surface deswelling of thermoresponsive microgels by electrophoresis. Langmuir 2022, 38, 16084-16093.

https://doi.org/10.1021/acs.langmuir.2c02742

5.

The authors are recommended to apply their method for real thermoresponsive gels, that is, they should make careful literature search and select published experimental data, and compare with their model’s results. Alternatively, they have to prepare and measure the equilibrium swelling ratio (or swelling degree or swelling stretch) of thermoresponsive gels with well-defined crosslinking density as a function of temperature and analyze these data with the application of their approach.

6.

It is suggested that the authors should use swelling degree (or swelling ratio) and not the longitudinal “swelling stretch” for real thermoresponsive gels, because in most of the cases swelling ratio or swelling degree is measured as a function of temperature. See for instance the references in Comment 4 and references therein.

Author Response

Please find in attached.

Round 2

Reviewer 1 Report

Comments and Suggestions for Authors

Minor revision:

I appreciate all the revisions made by the authors. I further ask the authors to add the references for each data point (already presented in the SI) so that the datasets will become more reader-friendly if anyone in the future wants to check and use the dataets.

Reviewer 2 Report

Comments and Suggestions for Authors

This can be published in its present form.

Author Response

Thanks for your comments